# Finite Element Analysis Model of Electronic Skin Based on Surface Acoustic Wave Sensor

**DOI:** 10.3390/nano13030465

**Published:** 2023-01-23

**Authors:** Chunxiao Jiao, Chengkai Wang, Meng Wang, Jinghong Pan, Chao Gao, Qi Wang

**Affiliations:** 1College of Sciences, Northeastern University, Shenyang 110819, China; 2School of Materials Science and Engineering, Northeastern University, Shenyang 110819, China

**Keywords:** electronic skin, surface acoustic wave sensor, finite element simulation

## Abstract

In recent years, with the rapid development of flexible electronic devices, researchers have a great interest in the research of electronic skin (e-skin). Traditional e-skin, which is made of rigid integrated circuit chips, not only limits the overall flexibility, but also consumes a lot of power and poses certain security risks to the human body. In this paper, a wireless passive e-skin is designed based on the surface acoustic wave sensor (SAWS) of lithium niobate piezoelectric film. The e-skin has the advantages of small size, high precision, low power consumption, and good flexibility. With the multi-sensing function of stress, temperature, and sweat ion concentration, etc., the newly designed e-skin is a sensor platform for a wide range of external stimuli, and the measurement results can be directly presented in frequency. In order to explore the characteristic parameters and various application scenarios of the SAWS, finite element analysis is carried out using the simulation software; the relationship between the SAWS and various influencing factors is explored, and the related performance curve is obtained. These simulation results provide important reference and experimental guidance for the design and preparation of SAW e-skin.

## 1. Introduction

E-skin generally consists of three parts: flexible stretchable sensors [1], electronic circuits [2], and skin-compatible patches [3,4]. It can be used for body shape management, motion capture, biophysical tracking [5], and even clinical analysis [4,6]. At present, research on the principle, type, structure, and application of e-skin is developing rapidly, and some technologies have been relatively mature. Various kinds of sensors can be applied to e-skin to achieve a variety of functions. For example, the strain resistance sensor based on carbon nanoparticles can be effectively applied to human motion detection [7] and the biosensors based on organic electrochemical transistors can be used to detect protein biomarkers of breast cancer effectively [8]. However, one limitation for the current widespread use of e-skins is their wireless, passive transmission. Traditional e-skin relies on rigid integrated circuit chips, such as Bluetooth, near field communication (NFC) or radio frequency identification (RFID) chips, and a microprocessor to achieve wireless transmission [9]. The reliance on these rigid integrated circuit chips has limited the overall flexibility and elasticity of the e-skins. In addition, the high power consumption of these chips usually leads to reduced device sensitivity [10], and the large amount of heat generated thereby [11] poses certain safety risks to the human body.

Under the background of the rapid development of fifth generation (5G) technology, the advantages of surface acoustic wave (SAW) have become increasingly prominent in the field of wireless passive signal transmission [12]. First of all, SAW is a kind of elastic wave propagating along the surface of objects. Its propagation speed is 4–5 orders of magnitude lower than that of electromagnetic waves, making it easy to introduce or extract signals in the propagation process. Secondly, SAW devices are similar to electromagnetic wave devices in that their size is comparable to the wavelength of SAW signals. The extremely low propagation speed and short wavelength of SAW give SAW devices the advantages of small size, low power consumption, stable performance, and good repeatability. Thirdly, SAW utilizes elastic waves on the crystal surface without involving the migration process of electrons. Therefore, SAW devices have strong anti-radiation ability and are not affected by electromagnetic waves. Fourthly, SAW devices are usually manufactured by semiconductor planar processes, which allow for easy mass production. At present, there is abundant research on SAWS, and their performance is often analyzed and evaluated by various modeling methods before preparation. These include, for example, the following: using an artificial neural network to model and analyze the frequency-dependent performance of SAW resonators [13]; using the direct and reliable equivalent-circuit extraction method to evaluate the performance of commercial SAW resonators [14]; and using finite element analysis and the model of coupling of modes based on P-matrices to calculate and measure the frequency response of SAW devices [15]. SAW devices have been widely used in stress and strain sensors, temperature and humidity sensors, biochemical sensors and many other fields [16,17,18,19,20,21,22].

In this paper, a wireless passive e-skin based on SAW sensors is envisaged. The idea is realized by integrating lithium niobate piezoelectric film onto flexible patches as wireless passive sensing material. Compared with traditional e-skin, it has greater flexibility, ductility, and elasticity, and can adapt to different working environments and meet the requirements of different scenarios to a certain extent. Finite element multi-physics simulation software simulates physical phenomena in real scenarios by coupling multiple physical fields. Its simulation model can be used as an important reference for experimental design, and the simulation results can be used to verify the experimental results [23]. On such a basis, a two-dimensional finite element simulation analysis model of SAW e-skin is constructed.

In this paper, the periodic structural elements of the SAW e-skin are analyzed by finite element simulation. The characteristic frequency, admittance, electromechanical coupling coefficient, *S11* parameter, and mechanical quality factor of the SAWS under initial conditions are obtained. Finally, the characteristic frequency changes of the SAWS model in different application scenarios such as stress, temperature, and sweat ion concentration are studied, respectively, and the relationship between the characteristic frequency and various influencing factors is summarized.

## 2. Materials and Methods

### 2.1. SAWS Principle

As shown in Figure 1, the SAWS is mainly composed of a piezoelectric substrate, an interleaved interdigital transducer (IDT) made of metal films fabricated on the polished surface of the substrate, and metal reflection grids on both sides.

As the excitation element of SAW, the IDT is installed on the piezoelectric substrate. If a high-frequency alternating electrical signal is added to both ends of the IDT electrode, the surface of the piezoelectric material will generate mechanical vibration due to the inverse piezoelectric effect and excite the SAW with the same frequency as the external electrical signal, which will propagate along the surface of the substrate. If a reflective grid is installed on both sides of the piezoelectric substrate, the mechanical wave will be reflected back to the IDT. The IDT uses the positive piezoelectric effect to convert mechanical waves into electrical signals and transmit them to the receiver. In this way, the measured information will be reflected by the changes in the speed or frequency of the SAWS [24].

### 2.2. Finite Element Simulation Model

#### 2.2.1. E-Skin Model

In SAW wireless passive e-skin, the semiconductor sensor, chip, and circuit element are replaced by the lithium niobate SAWS, and the changes in its resonance frequency can perfectly express the information about mechanical, temperature, humidity, and biochemical stimuli. Relevant toxicological data indicate that lithium niobate may cause minor skin irritation when it comes into contact with the skin. Therefore, surface-charge lithography [25] is used to coat lithium niobate with a layer of polydimethylsiloxane (PDMS; ~20 μm), and ventilation and cleaning are achieved through perforation [26]. As a non-toxic and non-irritating material to skin and mucous membrane, PDMS can effectively avoid contact between lithium niobate piezoelectric film and human skin. Meanwhile, PDMS has electrical insulation, mechanical elasticity [9,27], gas permeability, and good biocompatibility [9,28], which is an important material for the preparation of e-skin. Then the electrodes are deposited onto the lithium niobate piezoelectric film by lithography, and the IDT is connected to two flexible coils, which respectively input and output electrical signals. After the above process, the SAW e-skin is prepared. Compared with traditional e-skin, the SAW e-skin uses flexible SAWS instead of a rigid integrated circuit chip, uses SAW detection to reduce the potential damage to the skin caused by current detection, and uses perforated PDMS as a substrate. These measures improve the flexibility, safety, biocompatibility, and permeability of the e-skin [9,27,28], realize the wireless passive information transmission function, and save a lot of power.

#### 2.2.2. SAWS Simulation Model

The size design and performance calculation of SAW devices are complicated. The finite element method can improve the simulation accuracy and reduce the calculation time [29]. Therefore, COMSOL Multiphysics 5.6 finite element analysis software is adopted in this study.

As shown in Figure 2a, a periodic element is intercepted from the three-dimensional SAWS and is gradually simplified into a two-dimensional model. Relevant parameters of the two-dimensional finite element model are shown in Figure 2b. From top to bottom, the materials of each layer are aluminum electrode (IDT) and lithium niobate single crystal (piezoelectric layer). In Figure 2b, *λ* is the length of the resonant wave, *a* is the width of the electrode, *b* is the electrode spacing, *P* is the half wavelength, *D* represents one periodic unit, *h_Al_* is the thickness of the aluminum electrode, and *h_LN_* is the thickness of lithium niobate. Figure 2c shows the actual modeling of the SAWS in the finite element simulation software. Figure 2d shows the potential distribution of the model at the resonant frequency under the initial state. The material parameters, structural parameters, boundary conditions, and electrode polarity of each layer are presented in Table 1, Table 2, Table 3, Table 4 and Table 5.

## 3. Parameter Characteristics of SAWS

The admittance curve of the SAWS under initial conditions is shown in Figure 3a. Admittance is the amount of energy released by the system when the voltage changes, and is used to describe the difficulty for alternating current to pass through the circuit or system [30]. When the SAWS works, there are two characteristic frequencies under the condition of short circuit, namely resonant frequency and anti-resonant frequency. The left and right illustrations in Figure 3a show the deformed shape of the SAWS in resonant and anti-resonant states, respectively. As shown in Figure 3a, the resonant frequency of the SAWS is 1.8873 GHz, and the anti-resonant frequency is 1.9589 GHz. The relationship between SAW resonant frequency and phase velocity is shown in Formula (1):(1)v=f⋅λ

Accordingly, it can be obtained that the phase velocity of the SAW sensor in the initial state is 3774.6 m/s.

The *S* parameters (scattering parameters) describe the frequency domain characteristics of the transmission channel and reflect the relationship between the incident wave power and the reflected wave power of the network system. *S* parameters are important parameters to study the performance of the SAW resonator. According to different signal acquisition methods and ports, *S* parameters can be divided into *S11*, *S12*, *S21*, and *S22*. The performance evaluation of a single port resonator mainly depends on its *S11* parameter curve. The reflection coefficient of the resonator can be qualitatively analyzed based on the smoothness of the *S11* parameter curve. The smoother the curve, the greater the reflection coefficient of the resonator, and the better the performance of the resonator [31]. The *S11* curve of the resonant cavity of the SAWS and its reflection coefficient meet the following relationship [32]:(2)$$S11=\frac{V_{1}^{-}}{V_{1}^{+}},\;P=\frac{V^{2}}{R}$$
(3)$$\frac{P_{R}}{P_{I}}\left(\mathrm{dB}\right)=10\mathrm{lg}\frac{P_{R}}{P_{I}}$$
(4)$$S11\left(\mathrm{dB}\right)=10\mathrm{lg}S11^{2}=20\mathrm{lg}S11$$
where V is the voltage, R is the resistance value, $V_{1}^{-}$ is the voltage reflected signal at the port, $V_{1}^{+}$ is the voltage input signal at the port, P is the power, $P_{R}$ is the reflected power, and $P_{I}$ is the incident power. According to the above formulas, the greater the amplitude of the *S11* parameter curve of the device, the higher the reflection coefficient of the device. Figure 3b shows the *S11* parameter curve of SAWS under initial parameter conditions. From the figure, it can be seen that the SAW sensor has a good response around 1.8873 GHz, and the overall curve is smooth, without side lobes, and the amplitude can reach –100 dB.

Electromechanical coupling coefficient ($K^{2}$) represents the degree of mutual transformation between mechanical energy and electric energy during the vibration of piezoelectric vibrator, and is an important physical quantity to measure the piezoelectric strength of piezoelectric materials. Based on the finite element simulation, the electromechanical coupling coefficient can be expressed according to the relative interval between the resonant frequency and the anti-resonant frequency, as shown in Formula (5) [33]:(5)$$K^{2}=\left(\frac{\pi^{2}}{4}\right)\left(\frac{f_{m_{-}}-f_{m+}}{f_{m_{-}}}\right)$$
where $f_{m+}$ is the resonant frequency, and $f_{m_{-}}$ is the anti-resonant frequency. Based on this, the electromechanical coupling coefficients of the SAW sensor resonator under the condition of metal electrodes with different thickness can be calculated. Figure 3c depicts the variation curves of the electromechanical coupling coefficients with the thickness of the electrodes, when gold, copper, and aluminum are used as the electrodes. It can be seen from Figure 3c that aluminum, as the electrode, has a higher electromechanical coupling coefficient than copper and gold at the thickness range 0–0.25 μm, and a maximum value is reached at 0.2 μm. At the same time, aluminum is easier to be made into metal film and is relatively cheap, so 0.2 μm-thick aluminum is selected as the IDT electrode material.

The sensor loses a lot of energy in the wireless communication with the reader. This kind of loss is inevitable, and will directly affect the efficiency of wireless communication and further affect the transmission distance. In order to reduce the energy loss, it is necessary to improve the quality factor of the sensor terminal as much as possible. For single-port resonators, the relationship between the quality factor $Q_{m}$ and the device structure is shown in Formula (6) [34]:(6)$$Q_{m}=\frac{\pi L_{c}}{\lambda_{0}\left(1-\tan h_{LN}\left(\left|r_{s}\right|N_{g}\right)\right)}$$
where $L_{c}$ is the effective length of the resonant cavity, $r_{s}$ is the reflectivity of a single short-circuit electrode, and $N_{g}$ is the number of short-circuit electrodes in the reflection cavity.

Figure 3d shows the parameter curve of SAWS mechanical quality factor $Q_{m}$ under the initial structural parameter condition. From the figure, it can be seen that the SAWS has a good response near 1.8873 GHz, with a smooth curve and no sidelobe, and the amplitude can reach 10^21^.

When a computer is used for the simulation analysis, the influence of noise generated by the external electric field on simulation results cannot be ignored. In order to ensure the accuracy of simulation results, the noise immunity test is carried out. The initial input voltage value of the SAWS simulation model is shown in Table 5: electrode 1 is input 1 V and electrode 2 is grounded. Maintain the grounded state of electrode 2, change the input voltage of electrode 1, and record the resonant frequency. After the above steps, the relation curve between the input voltage of electrode 1 and the resonant frequency of SAW can be obtained, as shown in Figure 4a. Then change the relative state of electrode 1 and electrode 2, maintain the grounded state of electrode 1, change the input voltage of electrode 2, and record the resonant frequency in order to eliminate the influence of the external electric field on the electrode position. After the above steps, the relation curve between the input voltage of electrode 2 and the resonant frequency of SAW can be obtained, as shown in Figure 4b.

As shown in Figure 4, the simulation model of SAWS has good noise immunity performance, and the influence of the external electric field on simulation results can be ignored.

## 4. Application

As a sensing platform for a wide range of external stimuli, SAW e-skin has multiple sensing functions. By using the control variable method, the application scenarios of the e-skin including stress and strain, temperature, and sweat ion concentration are simulated and analyzed separately through the finite element method, and the measured results are directly presented through frequency.

### 4.1. Stress and Strain

SAW e-skin is very sensitive to stress and strain signals, so it can be used to capture and analyze human motion behavior, and can also be used to monitor human pulse and heartbeat in real time [9].

The elastic modulus of different materials is different. A change in external forces will lead to the deformation of the SAWS surface, which in turn will change the characteristic frequency or frequency response characteristics of the structure. Therefore, it is necessary to adopt “characteristic frequency of prestress” or “prestress and frequency domain” for analysis in finite element simulation.

The finite element stress simulation analysis of SAW is shown in Figure 5. Figure 5a is the schematic diagram of the external boundary load applied to the model surface. Figure 5b shows the stress diagram when the model is in resonant state after applying external boundary load. Figure 5c shows the comparison of admittance curves before and after stress application. On the left is the admittance curve when 10^8^ Pa stress is applied. On the right is the admittance curve with no stress applied. Figure 5d reflects the variation curve of the SAW resonant frequency caused by stress.

### 4.2. Temperature

Body temperature is one of the four vital signs of the human body and plays a very important role in the human body [35]. Human body temperature is affected by the thermoregulatory center to maintain body temperature within a certain range. Too high or too low body temperature will have a certain impact on the body. Many diseases, such as the common flu and the current pandemic of COVID-19, can lead to body temperature disorder [36]. Therefore, it is of great significance to monitor the temperature of the human body.

The influence of temperature on device performance is generally reflected in two aspects: the change of material properties by temperature and the influence of thermal stress generated by temperature on the device. The change of material properties caused by temperature is reflected in the fact that the properties of piezoelectric materials are related to temperature. It is only necessary to write the elastic matrix, coupling matrix, and dielectric matrix of piezoelectric materials as a function of temperature in the simulation model. The influence of thermal stress on the device is reflected in the different thermal expansion coefficients of different materials. The thermal stress effect generated by temperature changes will alter the characteristic frequency or frequency response characteristics of the structure. At this time, it is necessary to carry out the “prestressed characteristic frequency” or “prestressed and frequency domain” analysis in finite element simulation.

In this paper, lithium niobate is used as the piezoelectric substrate. As a high temperature ferroelectric material, the Curie temperature of lithium niobate is up to 1210 °C [37]. Therefore, temperature has little influence on the properties of lithium niobate materials, and the thermal stress generated by temperature (that is, the thermal expansion effect) is the main factor affecting the device.

The finite element temperature simulation analysis of SAW is shown in Figure 6. Figure 6a shows the stress diagram of the model when it is in resonant state under thermal expansion. Figure 6b is a comparison of admittance curves before and after thermal expansion caused by temperature change, with the admittance curve at 393.15 K on the left and the admittance curve at 293.15 K on the right. Figure 6c shows the variation curve of SAW resonant frequency with temperature. In order to meet the actual needs of e-skin temperature measurement, a temperature range of 298.15–318.15 K is selected for the study. It can be seen from the figure that the SAW resonance frequency decreases steadily with the increase of temperature.

### 4.3. Concentration

As an integral part of the human body, sweat is mainly composed of water, along with trace metal ions such as sodium, potassium, magnesium, and calcium [38]. According to relevant studies, the concentration of metal ions in sweat is closely related to some diseases [39]. Therefore, using e-skin to detect the concentration of related metal ions in sweat can provide an early warning for human health and an important reference for clinical diagnosis.

A planar diagram of sweat metal ion concentration measurement using the SAW e-skin device is shown in Figure 7a. In this paper, microfluidic technology [40] is innovatively combined with SAW sensing technology to obtain the microfluidic e-skin, which consists of four micro-reaction pools, micro-pumps, micro-valves based on ion-selective membranes preparation, micro-channels, and detection units based on SAWS. The micro-reaction pools are located at the four corners of the e-skin, and the bottom is made of material that can easily absorb sweat. When a certain amount of sweat is stored in the micro-reaction pools, the sweat is transmitted to the detection unit based on the SAWS through the micro-channels under the action of the micro-pump and micro-valve. When the sweat flows to the detection unit, the sweat will undergo solid–liquid separation under the action of the ion-selective membrane to screen out specific metal ions and form ion accumulation in the pipe between the IDT electrodes. The height of the ion accumulation layer is an important indicator of the concentration of the metal ions in sweat. Figure 7b presents a two-dimensional model for measuring sweat metal ion concentration using SAWS and the realization of changes in the height of the ion accumulation layer. As shown in Figure 7c, the influence of magnesium, calcium, and sodium ions on the SAW resonance frequency increases with the increase of the accumulation thickness, while the influence of potassium ions on the SAW resonant frequency decreases with the increase of accumulation thickness. Based on this, the accumulation thickness of the metal ions in the pipe between the IDT electrodes can be obtained by measuring the SAW resonance frequency under the action of different metal ions, and then the ion concentration of the metal ions in sweat can be calculated. In addition, microfluidic technology makes it possible to measure multiple metal ions simultaneously under the action of different ion-selective membranes. Furthermore, the SAW will produce different frequency offset responses to different metal ions in the propagation path, thus providing a new way to measure the composition and proportion of ions in sweat.

## 5. Conclusions

In this paper, a finite element analysis model for e-skin preparation based on SAWS is proposed, and detailed simulation studies are carried out from two aspects: the parameter characteristics and the application analysis of the model. Firstly, the characteristics of the resonant frequency, phase velocity, admittance, electromechanical coupling coefficient, *S11* parameter, and mechanical quality factor of the sensor under initial structural parameters are analyzed. The simulation results show that under initial structural parameters, the resonant frequency of the sensor is 1.8873 GHz, the phase velocity is 3774.6 m/s, the maximum electromechanical coupling coefficient occurs when 0.2 μm thick aluminum is used as the metal electrode, the *S11* parameter of the sensor can reach −100 dB, and the quality factor $Q_{m}$ is 10^21^. Secondly, in view of the fact that traditional e-skin, which relies on rigid integrated circuit chips, has limitations in overall flexibility, elasticity, and safety, a new e-skin based on SAW is studied, and its application feasibility in different scenarios is analyzed by finite element simulation software. The simulation results show that the resonant frequency of the SAWS is negatively correlated with stress when the SAW e-skin is under stress. When the ambient temperature changes, the SAW e-skin will be affected by thermal stress, and the resonant frequency of the SAWS will be offset and negatively correlated with temperature. Combined with microfluidic technology, the SAW e-skin can be used to measure the concentration of metal ions in sweat. In measuring the concentration of potassium, sodium, calcium, and magnesium in sweat, the resonance frequency of the SAWS is related to the thickness of ion accumulation to a certain extent, and then the correlation between the resonance frequency of the SAWS and the concentration of metal ions in sweat is obtained. In the above application scenarios, the relationship between the influencing factors and the resonant frequency of the SAWS is obvious, fully proving that the SAW e-skin has good working performance.

However, there are some limitations to the simulation analysis of the SAW e-skin. For example, when multiple application scenarios are applied at the same time, due to the potential cross-sensitivity among various influencing factors (e.g., when SAW e-skin is used to measure the concentration of metal ions in sweat, temperature and humidity may have a potential impact on the detection results; because it is affected by cross-sensitivity among ions, it is not clear how to evaluate the concentration of different ions by observing the movement of a single resonance frequency), the change of resonance frequency is not a simple superposition, and a certain factor may dominate in a specific environment. At present, we have not summarized the effective law of the resonant frequency change caused by multiple factors. In addition, under the two-dimensional simulation model, further discussion on the influence of various factors on the SAW resonance frequency is limited (e.g., when discussing the influence of changing relative humidity on SAW e-skin temperature measurement, we need to build a more complex temperature gradient or heat and humidity transfer model). These limitations provide new ideas for future work.

The research on SAW e-skin is not limited to this, but can be expanded in multiple directions, such as researching other piezoelectric materials, designing multi-layer structures to achieve better SAW propagation characteristics, and exploring the influence of humidity, torque, acid, alkaline, and other external conditions on the performance of SAW e-skin. In future work and research, the authors will prepare the SAW e-skin according to the simulation results of this paper, test the simulation model designed in this paper, and compare the performance of the SAW e-skin with that of traditional e-skin more clearly, so as to promote the development and application of the SAW e-skin.

## Figures and Tables

**Figure 1 nanomaterials-13-00465-f001:**
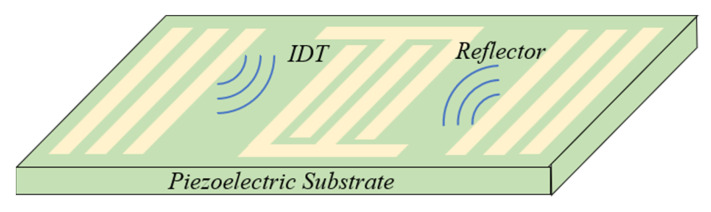
Schematic diagram of the SAWS.

**Figure 2 nanomaterials-13-00465-f002:**
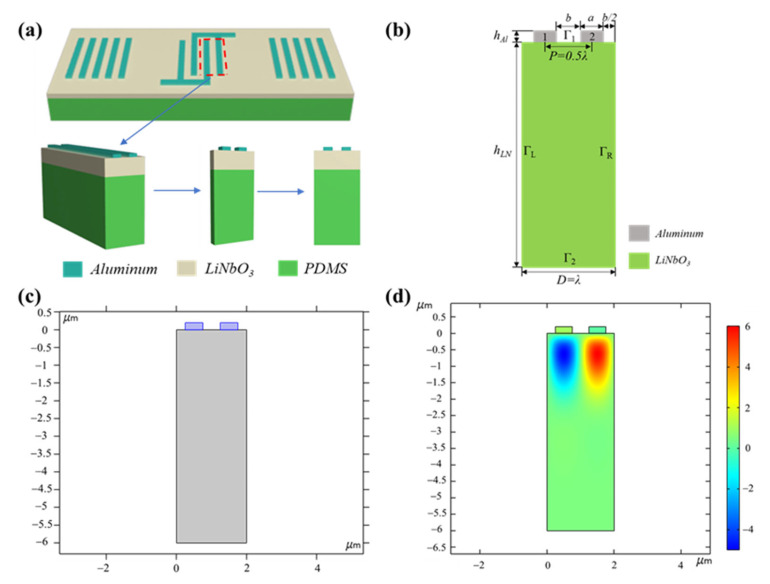
Simulation modeling process for SAWS. (**a**) Selection of periodic element for the two-dimensional model of SAWS; (**b**) two-dimensional finite element model construction and related parameters; (**c**) two-dimensional model of SAWS in finite element simulation; (**d**) potential distribution of SAW in simulation modeling.

**Figure 3 nanomaterials-13-00465-f003:**
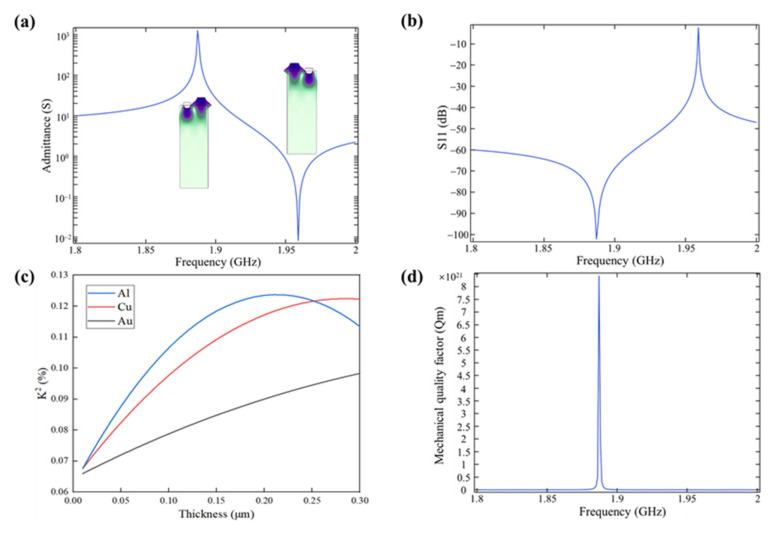
Parameter characteristics of the SAWS. (**a**) The admittance curve of the SAWS under the initial condition (the figure shows the deformed shape of the SAW when it is excited); (**b**) the *S11* parameter of the SAWS is expressed in decibels; (**c**) variation curve of electromechanical coupling coefficient with electrode thickness when gold, copper, and aluminum are used as electrodes; (**d**) $Q_{m}$ parameter curve of SAWS.

**Figure 4 nanomaterials-13-00465-f004:**
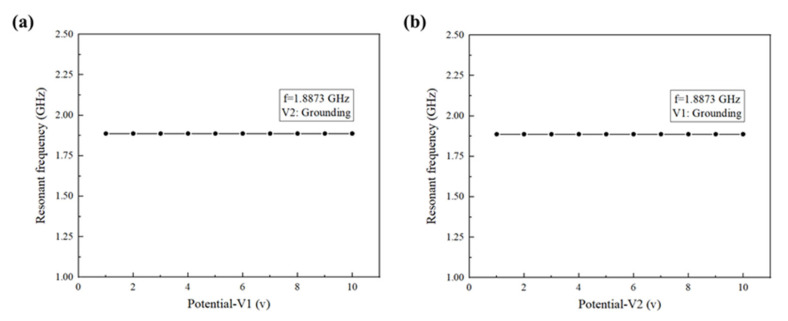
Noise immunity analysis of the SAWS. (**a**) The relation curve between the input voltage of electrode 1 and the resonant frequency of SAW; (**b**) the relation curve between the input voltage of electrode 2 and the resonant frequency of SAW.

**Figure 5 nanomaterials-13-00465-f005:**
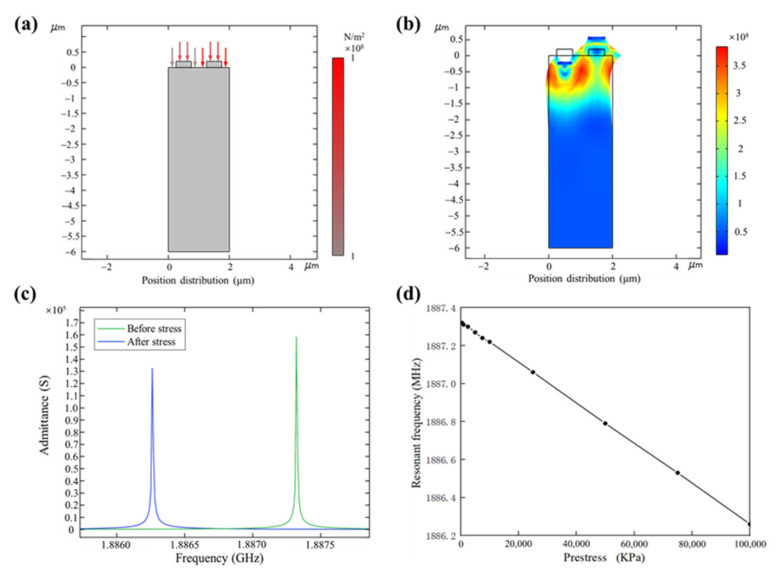
Finite element stress simulation analysis of SAW. (**a**) Schematic diagram of stress applied on the model surface; (**b**) schematic diagram of finite element simulation stress; (**c**) comparison of admittance curves before and after stress application; (**d**) the relationship between stress and the resonant frequency of SAW.

**Figure 6 nanomaterials-13-00465-f006:**
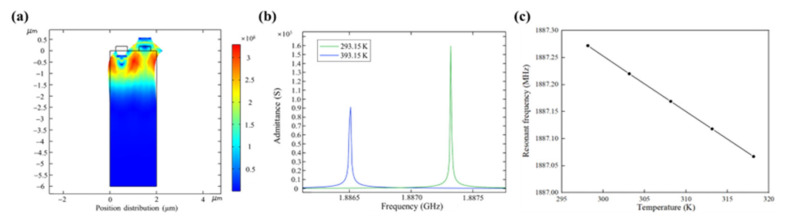
Finite element temperature simulation analysis of SAW. (**a**) Stress diagram of finite element simulation model affected by temperature; (**b**) comparison of admittance curves before and after temperature change; (**c**) the relation curve between temperature and resonant frequency of SAW.

**Figure 7 nanomaterials-13-00465-f007:**
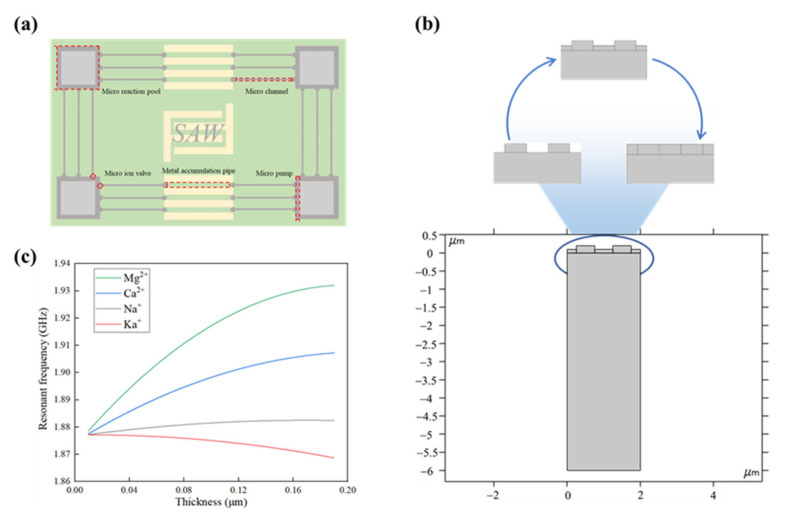
Sweat ion concentration measurement using SAWS. (**a**) Schematic diagram of extracting metal ions from sweat using microfluidic techniques and ion-selective membranes; (**b**) finite element simulation model for measuring sweat ion concentration using SAW; (**c**) the relationship between the resonant frequency of SAW and the thickness of ion accumulation.

**Table 1 nanomaterials-13-00465-t001:** Initial structural parameters of the simulation model.

Name	Parameter
*λ*	2 μm
*a*	0.25*λ* (0.5 μm)
*b*	0.25*λ* (0.5 μm)
*P*	0.5*λ* (1 μm)
*D*	*λ* (2 μm)
*h_Al_*	0.2 μm
*h_LN_*	6 μm

**Table 2 nanomaterials-13-00465-t002:** Parameters of piezoelectric materials in the simulation model.

Material	128° Y-X LiNbO_3_	
Density, *ρ* [kg/m^3^]	4700	
Relative dielectric constant, *ε_rs_*	{epsilonrS11, epsilonrS22, epsilonrS33}; epsilonrSij = 0	{43.6, 43.6, 29.16}
Coupling matrix, *e_ES_* [C/m^2^]	{eES11, eES21, eES31, eES12, eES22, eES32, eES13, eES23, eES33, eES14, eES24, eES34, eES15, eES25, eES35, eES16, eES26, eES36}	{0, −2.53764, 0.193644, 0, 2.53764, 0.193644, 0, 0, 1.30863, 0, 3.69548, 0, 3.69594, 0, 0, −2.53384, 0, 0}
Elastic matrix, *C_E_* [10^11^ Pa]	{cE11, cE12, cE22, cE13, cE23, cE33, cE14, cE24, cE34, cE44, cE15, cE25, cE35, cE45, cE55, cE16, cE26, cE36, cE46, cE56, cE66}; cEij = cEji	{2.02897, 0.529177, 2.02897, 0.749098, 0.749098, 2.43075, 0.0899874, −0.0899874, 0, 0.599034, 0, 0, 0, 0, 0.599018, 0, 0, 0, 0, 0.0898526, 0.748772}

**Table 3 nanomaterials-13-00465-t003:** Material parameters of IDT in the simulation model.

Material	Al (IDT)
Density, *ρ* [kg/m^3^]	2700
Young’s modulus, *E* [Pa]	70 × 10^9^
Poisson’s ratio, *µ*	0.33

**Table 4 nanomaterials-13-00465-t004:** Boundary conditions used in the simulation model.

Boundary	Mechanical Conditions	Electrical Conditions
Γ_1_	Free	Zero charge
Γ_2_	Fixed	Ground
Γ_L_, Γ_R_	Periodic boundary conditions	Periodic boundary conditions

**Table 5 nanomaterials-13-00465-t005:** Electrode polarity of IDT in the simulation model.

Electrode Number	Al (IDT)
1	+1 V
2	Grounding

## Data Availability

All available data is contained within the article.

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
