# Peer review of "Finite Element Analysis Model of Electronic Skin Based on Surface Acoustic Wave Sensor"

_nanomaterials, 2023, doi:10.3390/nano13030465_

Round 1
Reviewer 1 Report
The manuscript "Finite element analysis model of electronic skin based on surface acoustic wave sensor" presents the current and future development of electronic skin based on the SAW mechanism. The results of modeling the response of the sensor to mechanical, temperature effects, as well as to the effects of various concentrations of ions in sweat are described in detail. However, the numerical advantages of the presented sensor in the context of other developments in this topic are not clearly indicated.
Several shortcomings have been identified in the manuscript, the improvement of which will significantly improve the overall quality and accessibility of the manuscript.
1. Figure 1 and Figure 2 are almost identical. Figure 2 has no informational value, perhaps Figure 2 should be replaced or removed.
2. In Lines 91-93 states that, compared with traditional e-skin, electronic skin based on surface acoustic wave sensor has better flexibility, safety, biocompatibility and durability. This statement must be supported by experimental comparative studies. Such a conclusion cannot be drawn from simulations.
3. Please, explain in more detail the designations in Table 1. In the current version, the meaning of the parameters is not obvious.
4. In figures 5c and 6b, it is worth signing the corresponding curves before and after mechanical loading.
5. It would be useful to mention studies on the toxicity of lithium niobate piezoelectric film and possible precautions as the material is used in contact with human skin.
6. The authors mention the good flexibility of the developed e-skin, but it remains unclear how the flexibility was evaluated and how the degree of "good" is quantified. Please add a comparison and explanation to the declared parameters.
7. Line 89. Please explain how lithium niobate and metal electrodes were attached to polydimethylsiloxane. What does the minus sign in "-20 µm" mean? Perhaps it meant the thickness of the substrate?
8. Please explain how and with what it is planned to attach the test device to human skin to monitor the declared parameters (deformation, temperature, concentration of sweat ions).
9. Please check the X-axis labels in Figure 5 a,b and Figure 6a.
10. As part of the work, the noise immunity of the SAW e-skin sensor is not sufficiently disclosed. In silico studies were carried out on the ability of the SAW sensor to register external influences, but the effects of external electric fields were not taken into account.
11. The proposed version of e-skin involves reading several parameters at once: mechanical load, temperature, concentration of ions in sweat due to the study of the resonant frequency. Please explain how the determination of which particular parameter under study has led to a change in the resonant frequency is made.
12. What determines the competitiveness of this device? There is a large number of works on the creation of economical sensors based on carbon nanotubes and graphene. Adding to the Introduction section an overview of current research in the field of flexible sensors for monitoring biological characteristics will emphasize the importance and relevance of the study. Particularly relevant is the technology of manufacturing strain-resistive sensors based on carbon nanoparticles, for example: Demidenko, N.A.; Kuksin, A.V.; Molodykh, V.V.; Pyankov, E.S.; Ichkitidze, L.P.; Zaborova, V.A.; Tsymbal, A.A.; Tkachenko, S.A.; Shafaei, H.; Diachkova, E.; Gerasimenko, A.Y. Flexible Strain-Sensitive Silicone-CNT Sensor for Human Motion Detection. Bioengineering 2022, 9, 36. https://doi.org/10.3390/bioengineering9010036
Reviewer 2 Report
Overall, the manuscript is well-written and presents a thorough analysis of the design and performance of a wireless passive electronic skin (e-skin) based on surface acoustic wave sensors (SAWS). The authors have provided a detailed description of the materials and methods used in the design and simulation of the SAW e-skin, as well as a thorough analysis of its characteristics and potential applications. However, before the manuscript can be considered for publication in the Nanomaterials journal, the authors are encouraged to address the following comments:
1) In the Introduction section, it would be helpful to provide more background information on the surface acoustic wave sensor (SAWS) with special attention to the methods used to model them, such as artificial neural networks (ANNs) [1], equivalent circuit models [2], and COM based P-matrices [3].
[1] Z. Marinković, G. Gugliandolo, G. Campobello, G. Crupi and N. Donato, "Application of Artificial Neural Networks for Modeling of the Frequency-Dependent Performance of Surface Acoustic Wave Resonators," 2021 56th International Scientific Conference on Information, Communication and Energy Systems and Technologies (ICEST), 2021, pp. 145-148, doi: 10.1109/ICEST52640.2021.9483548.
[2] Gugliandolo, G.; Marinković, Z.; Campobello, G.; Crupi, G.; Donato, N. On the Performance Evaluation of Commercial SAW Resonators by Means of a Direct and Reliable Equivalent-Circuit Extraction. Micromachines 2021, 12, 303. https://doi.org/10.3390/mi12030303
[3] Koigerov, A.S. Modern Physical-Mathematical Models and Methods for Design Surface Acoustic Wave Devices: COM Based P-Matrices and FEM in COMSOL. Mathematics 2022, 10, 4353. https://doi.org/10.3390/math10224353
2) In the Application section, it would be beneficial to provide more information on the performance of the e-skin under different conditions, such as varying relative humidity. This would help to demonstrate the robustness and versatility of the e-skin design in temperature measurements.
3) In subsection 4.3 the authors write "In addition, microfluidic technology makes it possible to measure multiple metal ions simultaneously under the action of different ion-selective membranes. What’s more, the SAW will produce different frequency offset responses to different metal ions in the propagation path, thus providing a new idea to measure the composition and proportion of ions in sweat." It is not clear how it is possible to evaluate the concentration of different ions by observing the shift of a single resonant frequency. The authors should clarify this point and also comment on the potential cross-sensitivity of temperature in metal ion detection.
4) There are also a few typos that need to be addressed:
Line 47: "Fourthly, SAW devices are usually manufactured by semiconductor planar process, which make it easy to be mass produced." "Processes"? Or "makes"?
Table 3: I would suggest writing "70 x 10^9" (using superscript for the exponent) instead of "70e9".
Line 131: "The S-parameter (scattering coefficient) describes..." Which S-parameter? Do you mean "The S-parameters"? Moreover, the term "S parameters" should be consistently written throughout the manuscript: please choose "S-parameters" or "S parameters".
Round 2
Reviewer 1 Report
Dear Editor,
I believe that the article can be published.